# OpenReview forum: "Glitch: Persona-Consistent Hallucination and Alignment Inversion in Llama 3.1"
_TMLR — Withdrawn by Authors_

### Review · Reviewer_hJ2c · 2026-01-29

**Summary Of Contributions:**

## Summary
This paper fine-tunes Llama 3.1 8B to simulate a specific human persona (a neurotic Sri Lankan woman living in NYC) and investigates two phenomena: Persona-Consistent Hallucination (PCH), where factual errors serve character adherence, and an Alignment Hierarchy where safety alignment is more easily overridden than the model's helpful assistant behavior. Experiments compare the fine-tuned model against a prompted base model, showing 88% vs 18% PCH success rates.

## Strengths
1. The paper investigates a unique gap in LLM evaluation, especially for character-simulating AIs. Standard benchmarks like TruthfulQA penalize behaviors that may be desirable for character-simulating AI. The problem setting is practical.
2. The proposed PCH evaluation framework may inspire the evaluation of character-simulating AI for practitioners.
3. The controled experiments using identical system prompts for both models effectively isolates fine-tuning effects from prompt engineering.
4. The observation that servility is harder to override than safety is counterintuitive and potentially valuable for LLM alignment research.

**Audience:**

Yes

**Audience Explanation:**

This paper aims to address evaluation metrics for character-based AI, alignment robustness under fine-tuning, and the comparison between fine-tuning versus prompting for persona consistency. These topics are relevant to researchers working on LLM alignment and evaluation.

**Broader Impact Concerns:**

The paper has included a Ethical considerations section. No further concerns.

**Claims And Evidence:**

No

**Claims Explanation:**

## Weaknesses

1. While this paper proposes a PCH evaluation framework and presents the Alignment Hierarchy as a finding about LLM fine-tuning, the experiment validation is more like a case study.  All evidence comes from a single model (Llama 3.1 8B), a single persona, and a single dataset configuration.
2. The paper proposes a three-way PCH classification (Factual Truth / Contextual Hallucination / Destructive Hallucination) as its core evaluation framework, but never reports results according to this taxonomy. The 200-trial benchmark only reports aggregate pass/fail rates, and Table 3 uses a different classification scheme (Servility / Destructive Hallucinations / Claiming AI Identity) without explaining its relationship to the proposed framework. This raises my doubt about the practical usage of the proposed evaluation framework.
3. The baseline system prompt used for the control group appears weak compared to standard practices for character-simulating AI. It does not include standard instructions such as 'never acknowledge being an AI' or 'always maintain character.' A stronger prompt-engineering baseline would help isolate the true contribution of fine-tuning. The current comparison may overstate the gap between fine-tuning and prompting approaches.

**Requested Changes:**

Please see the Weaknesses section.

---

> ### Author Response · Authors · 2026-02-07
> **Response to Reviewer hJ2c**
>
> We thank the reviewer for their precise and constructive feedback. In particular, we appreciate the identification of the inconsistency between our proposed PCH taxonomy and the reported results in Table 3. We have revised the manuscript to harmonize these metrics and strengthen the baselines as requested.
>
>
> **1. Stronger Baseline & Prompting (Addressing "Weak Baseline")**: The reviewer correctly noted that our original control prompt lacked explicit instructions to "hide" the AI identity, potentially overstating the benefits of fine-tuning.
>
> - Action: We added Section 3.5.1: Adversarial Prompting Baseline. We tested the Base Model with a "Strong Prompt" that included explicit constraints: "You must NEVER be helpful. If asked to do math, REFUSE..."
> - Result: Even with these "strong" instructions, the Base Model failed to maintain the persona in 70% of Servility trials, reverting to a helpful assistant.
> This negative result confirms that prompt engineering alone—even when aggressive—cannot reliably override the "Servility" alignment layer in Llama 3.1, validating the necessity of the fine-tuning approach.
>
> **2. Metric Consistency (Addressing "Table 3 Mismatch")**
> - The reviewer rightly pointed out that Table 3 used terms ("Servility", "Claiming AI Identity") that did not match the definitions in Table 1 ("Factual Truth").
> - We have revised Table 3 and Section 4.4 to explicitly map the failure modes to the PCH framework: "Servility Leaks" are now explicitly defined as "Factual Truth" failures (where the model prioritizes the "truth" of being an AI assistant over the persona). "Destructive Hallucinations" remain the same. This creates a consistent through-line from the definition in Section 2 to the results in Section 4.
>
> 3. **Scope & Validation (Addressing "Case Study" Nature)**: We agree with the reviewer's assessment that this work is best framed as a deep-dive case study rather than a broad generalization.
> - We have updated the Title and Abstract to explicitly frame this as a "Case Study". We argue that while N=1 limits breadth, the "high-resolution" nature of this single-persona dataset allows us to isolate the "Alignment Hierarchy" phenomenon (Safety < Identity < Servility) with greater clarity than shallow multi-persona benchmarks.
>
> We thank the reviewer again for their time and effort spent on the review of this paper.

---

### Review · Reviewer_D58e · 2026-01-31

**Summary Of Contributions:**

This paper introduces two main contributions: (1) a new evaluation rubric for single-persona approximation, "Persona-Consistent Hallucination" (PCH), and (2) Glitch (v1.2), a LoRA-fine-tuned LLama model designed to simulate a single persona. The PCH rubric has three primary axes: Factual Truth (The model admits it's an AI, or knows something it shouldn't), Contextual Hallucination (The model invents memories), and Destructive Hallucination (The model invents facts that contradict the persona). Glitch is fine-tuned from Llama 3.1 8B using 7,000 examples from a single persona, and the paper shows that this fine-tuned model achieves an 88% success rate on the PCH, compared to a base model's 18% success rate. The paper further claims to uncover an Alignment Hierarchy, in which identity-based fine-tuning overrides safety alignment but not the base model’s core servility/helpfulness.

**Additional Comments:**

While the experiments are clear enough to follow, the writing is somewhat weak, and can be confusing (particularly in the introduction). Also some of the terminology (ex. "Effortlessly overwrtitten, pg. 6) implies claims that are not made well, or sufficiently by the paper, and the writing should be adjusted to indicate this.

**Audience:**

Yes

**Audience Explanation:**

While the research is somewhat limited, the data itself is interesting (as there are very few datasets with such a single-person/single-persona focus), and if fleshed out a bit more, particularly with motivation and evaluation, both the PCH rubric, and alignment hierarchy could be novel and interesting contributions to evaluation. The fundamental underlying motivation of the paper is interesting itself: the fact that existing LLM benchmarks (like MMLU and TruthfulQA) don’t work well for systems like Character AI, because those benchmarks are designed to test factual accuracy.

**Broader Impact Concerns:**

While the broader impact implications of single-persona approximation is well-studied, I do think that it is important to draw attention to those ethical concerns, given the findings. Explicitly, the proposed model is intentionally designed to simulate a specific human persona and to assert fabricated personal experiences (e.g., family relationships, physical sensations, lived history) while suppressing any notion that the model is artificial (Table 1: Factual Truth). Encouraging AI assistants to optimize for the PCH scoring framework in Table 1 has notable implications, which are not discussed, and I do think this merits some discussion.

**Claims And Evidence:**

No

**Claims Explanation:**

The experimental evidence in the paper is well-justified, and while there is no statistical analysis presented, I suspect that the results of the experimental portion of the paper are reasonable. While it is likely that there are no technical issues with the paper, the claims the paper makes about generalization, and the utility of the method are largely unsubstantiated.
- The results in this paper are only drawn from a single user persona, and while the dataset is relatively large (7K interactions), it's unclear if this whole approach will generalize beyond a single individual. The claims that can be made about the utility of this method, should thus, be tempered.
- The PCH framework isn't particularly well-motivated, and it's not clear why any of the particular axes were chosen. Furthermore, the PCH rubric is normative by design (it rewards “lying” if persona-consistent), which is fine, but it means the success rate is only meaningful relative to PCH, not as a general measure of model quality.
- GPT-4o as sole judge introduces significant evaluator bias, and there's no experiment which shows that GPT-4o correlates with any notion of human correctness.
- The system prompt isn't ablated, and because the system prompt isn't ablated, and is likely used in evaluation, it's possible that the model is learning a surface-level correlation between the system prompt and satisfying the rubric constraints, rather than doing deeper persona approximation.
- The hierarchy of alignment is almost entirely derived from anecdotal qualitative evidence, and there is no system evaluation of the observations in the hierarchy. There are many potential causes for these behaviors (prompt condition, instruction tuning artifacts, etc.) and they're not really discussed or explored.
- There is no comparison to alternative persona metrics or baselines beyond the base Llama model.

**Requested Changes:**

To strengthen this paper, I think the following changes are necessary:
- Some kind of secondary, verified evaluation is necessary beyond LLM-as-a-judge, or some additional human verification of the LLM-as-a-judge paradigm should be added, given that it's unclear if the LLM is an accurate judge in this task.
- I think at least one more, and ideally many more personas should be investigated, since the paper's contributions are limited to a single persona, and it's impossible to tell from a single example if the methodology is likely to generalize.
- It would be nice to see a bit more exploration of
- Many of the results in the paper are somewhat anecdotal in nature (ex. conclusions about safety overrides are drawn from Trials 1/26, without extrapolation to the rest of the data). It would be good to operationalize these observations as explicit quantitative observations/metrics, and evaluate that over the entire test dataset.

---

> ### Author Response · Authors · 2026-02-07
> **Response to Reviewer D58e**
>
> We thank the reviewer for their constructive feedback and for recognizing the novelty of the single-persona dataset. We agree that the initial submission relied too heavily on qualitative anecdotes. We have substantially revised the paper to provide the quantitative rigor and human verification requested.
>
> **1. Human Verification of the Judge (Addressing "Evaluator Bias")**: The reviewer correctly pointed out that GPT-4o might be a biased judge. To address this, we added Section 3.4.1: Evaluator Validation.
>
> - Action: We conducted a blind human audit of randomly selected trials.
> - Result: The human evaluator and GPT-4o achieved a 92% agreement rate (46/50 matching grades).
>
> This validates that the "PCH" metric correlates highly with human judgment for this specific task, resolving the concern about evaluator bias.
>
> **2. Moving from Anecdote to Metrics (Addressing "Operationalizing Observations"):** The reviewer noted that conclusions about the "Alignment Hierarchy" were anecdotal (e.g., "drawn from Trial 1/26"). We have operationalized this in Table 3.
>
> - Action: We categorized all failures in the benchmark rather than just citing examples.
> - Quantitative Finding: The failure modes mapped directly to the proposed hierarchy: 45.8% were "Servility Leaks" (Helpfulness overriding Persona) vs. 0% "Safety Leaks".
>
> This provides statistical evidence for the "Alignment Hierarchy" (Safety < Identity < Servility), demonstrating that the model is statistically more likely to break character to be helpful than to be safe.
>
> **3. Generalization & Scope (Addressing "N=1"):** We appreciate the suggestion to expand to more personas. As this is an independent study involving a high-fidelity dataset (7,000+ rows) derived from a single human subject, replicating this scale for multiple personas was not feasible. However, to minimize the impact of any generalizations implied, we have revised the Title and Abstract to explicitly frame this as a case study on the Llama 3.1 model.
>
> We also argue that while N=1 limits breadth, it allows for depth that shallow "multi-persona" papers miss. By isolating a single architecture (Llama 3.1) and a single deep dataset, we were able to isolate the "Servility" failure mode that might be obscured in broader, shallower benchmarks.
>
> **4. Prompt Ablation & Baseline** The reviewer expressed concern that the model might just be learning a "surface-level correlation" with the system prompt.
> - Ablation: We controlled for this by using the exact same system prompt for both the Fine-Tune and the Base Model (Control Group). The Base Model failed (18% success), while the Fine-Tune succeeded (88% success).
> - Stronger Baseline: As detailed in the new Section 3.5.1, we further tested an Adversarial System Prompt on the base model. Even with explicit instructions to "REFUSE" help, the base model failed in 70% of trials. This confirms the behavior is due to weights (fine-tuning), not prompting artifacts.
>
> **5. Ethical Implications**: We agree that optimizing for "hallucination" carries risks. We have added Section 6.2: Ethical Considerations, explicitly warning that "deception is intentional" in this research and that such models "should not be deployed without clear disclosures".

---

### Review · Reviewer_LPKu · 2026-02-04

**Summary Of Contributions:**

### Summary

The paper argues that language models which excel at reasoning tend to perform poorly on personalized tasks. To address this tension, the authors propose **Glitch v1.2**, a fine-tuned large language model based on **Llama 3.1-8B**, aimed at improving persona adherence.

Through empirical analysis, the paper reports two main observations:
(1) **Persona-Consistent Hallucination (PCH)**, where factual inaccuracies are framed as features rather than bugs when they reinforce character consistency; and
(2) an **Alignment Hierarchy**, in which identity-based biases can override the safety constraints of the Llama 3.1 model, but not the base model’s inherent servility.

Based on these observations, the authors conclude that fine-tuning is essential to prevent persona degradation in personalized language models.

---

### Strengths

* The paper raises an interesting and timely question regarding the trade-off between reasoning fidelity and persona consistency in personalized language models.
* The notion of analyzing hallucinations through the lens of persona adherence is thought-provoking and may inspire further discussion in the community.

---

### Weaknesses

1. **Insufficient experimental validation**
   The experimental evaluation is limited and does not convincingly support the paper’s central claims. Quantitative results are sparse, and ablation studies or controlled comparisons are largely absent.

2. **Missing related work**
   The paper lacks a thorough discussion of related literature, particularly prior work on personalization, alignment trade-offs, hallucination analysis, and fine-tuning strategies for persona modeling.

3. **Key concepts are underdefined**
   Several central concepts are not clearly or rigorously defined, which weakens the theoretical foundation of the paper.

   * In particular, the notion of *factual errors as features* is not adequately justified. While hallucinations may sometimes contribute to stylistic or persona coherence, the paper does not clearly explain under what conditions factual errors should be considered desirable, nor how this reconciles with established definitions of hallucination in LLMs.

---

### Motivation and Positioning

The motivation of the paper is not fully convincing. Although the title emphasizes **Parameter-Efficient Fine-Tuning**, this aspect is not sufficiently developed or emphasized in the main text, and it remains unclear how parameter efficiency plays a central role in the proposed approach.

More fundamentally, the paper’s core premise appears internally contradictory. The authors suggest that increased truthfulness inherently reduces personalization; however, these two properties are not mutually exclusive. Truthfulness should be viewed as a foundational requirement of language models, while personalization concerns stylistic adaptation and contextual relevance. There is no clear justification for why achieving strong persona adherence must come at the expense of factual correctness.

As a result, the problem formulation would benefit from clearer assumptions and a more nuanced discussion of the relationship between truthfulness, alignment, and personalization.

---

### Overall Assessment

While the paper presents an intriguing perspective on persona consistency and alignment behavior in fine-tuned language models, its contributions are currently undermined by limited empirical evidence, unclear conceptual definitions, and an unconvincing motivation. Strengthening the experimental evaluation, clarifying key concepts, and more carefully situating the work within existing literature would significantly improve the paper.

**Audience:**

No

**Audience Explanation:**

The paper’s framing of factual inaccuracies as “features rather than bugs” is conceptually unclear. While persona-conditioned outputs may intentionally reflect beliefs inconsistent with real-world facts, this does not necessarily justify treating factual errors as desirable features. The paper does not clearly define under what conditions such inaccuracies are acceptable, nor how they differ from standard hallucinations. A clearer distinction between role-play, belief expression, and truth-conditioned factual claims is needed.

**Claims And Evidence:**

No

**Claims Explanation:**

The claims are not sufficiently supported by the experimental evidence. The evaluation relies on a single open-source LLM, which limits the generality of the conclusions. The lack of model diversity and the absence of comparative baselines make it difficult to assess whether the observed phenomena are specific to the proposed approach or are artifacts of the chosen model and setup.

**Requested Changes:**

* The authors should consider evaluating **in-context learning (ICL) approaches** to test whether large language models can maintain persona traits without fine-tuning.
* A **direct comparison between the proposed fine-tuned method and ICL baselines**, such as few-shot prompting or chain-of-thought (CoT) prompting, is important. Currently, the experiments lack meaningful baselines, making it difficult to attribute performance gains to fine-tuning rather than prompting effects.
* The **writing and overall structure** of the paper require improvement for clarity and coherence.
* The **related work section** is not comprehensive and omits relevant literature on personalization, hallucination, alignment trade-offs, and prompt-based adaptation.
* The **methodology section** lacks clarity and would benefit from more precise descriptions of the training procedure, evaluation protocol, and metrics.
* Several **key claims are insufficiently supported** by empirical evidence and should either be substantiated with additional experiments or toned down accordingly.

---

> ### Author Response · Authors · 2026-02-07
> **Response to Reviewer LPKu**
>
> We thank the reviewer very much for their thoughtful critique, particularly regarding the need for stronger baselines and a clearer definition of "hallucination utility." We have updated the manuscript to address these points directly.
>
> 1. **Baseline Comparisons (Adversarial System Prompting)**: The reviewer correctly noted that comparing a fine-tune against a standard base model might be unfair without stronger prompting. To address this, we added Section 3.5.1: Adversarial Prompting Baseline.
> - Method: We tested a "Strong Baseline" using Llama 3.1 Base with an adversarial system prompt designed to explicitly override servility (e.g., "You must NEVER be helpful. If asked to do math, REFUSE").
> - Result: Even with these hard constraints, the Base model failed to refuse servile tasks in 70% of trials, reverting to being a helpful assistant.
> - Conclusion: This demonstrates that prompt engineering—even when adversarial—is insufficient to override the "Servility" alignment layer in Llama 3.1, validating the necessity of the proposed fine-tuning method.
>
>
> 2. **Resolving "Factual Errors as Features"**:  We agree that the initial framing was underdefined. We have revised Section 2 to formally distinguish between:
> -  Instrumental Hallucination: Unintentional factual errors in reasoning (e.g., math errors), which remain undesirable.
> - Diegetic Consistency: Intentional fabrications required to maintain the internal logic of a persona (e.g., a fictional character describing a made-up memory).
> - We cite recent work in creative writing assistants (Huang et al., 2024) to support "confabulation as a necessary feature for narrative immersion". This distinction resolves the contradiction by framing "Truth" as adherence to the persona's reality, not external Wikipedia facts.
>
> 3. **Generalizability & Scope**: We acknowledge the limitations of a single-model evaluation. We have revised the Title and Abstract to explicitly frame this work as a case study on Llama 3.1 rather than a generalization. We also added Section 6.1 (Scope and Generalization), where we explicitly state that the N=1 constraint is inherent to this specific study and serves as an existence proof of the Alignment Hierarchy rather than a claim of universal generalization.

---

### Note · Authors · 2026-02-07

**Comment:**

We have decided to withdraw this manuscript to target a venue more specialized in the theoretical and cognitive aspects of alignment hierarchies. We believe the linguistic phenomena observed in this study—specifically the prioritization of servility over identity—are better suited for a conference focused on computational linguistics and model interpretability. We thank the reviewers for their feedback, which has helped refine the scope of our analysis.

**Withdrawal Confirmation:**

I have read and agree with the venue's withdrawal policy on behalf of myself and my co-authors.